# Learning to Use Future Information in Simultaneous Translation

## Abstract

Simultaneous neural machine translation (briefly, NMT) has attracted much attention recently. In contrast to standard NMT, where the NMT system can access the full input sentence, simultaneous NMT is a prefix-to-prefix problem, where the system can only utilize the prefix of the input sentence and thus more uncertainty and difficulty are introduced to decoding. *Wait-k* (Ma et al., 2019) inference is a simple yet effective strategy for simultaneous NMT, where the decoder generates the output sequence $k$ words behind the input words. For *wait-k* inference, we observe that *wait-m* training with $m > k$ in simultaneous NMT (i.e., using more future information for training than inference) generally outperforms *wait-k* training. Based on this observation, we propose a method that automatically learns how much future information to use in training for simultaneous NMT. Specifically, we introduce a controller to adaptively select *wait-m* training strategies according to the network status of the translation model and current training sentence pairs, and the controller is jointly trained with the translation model through bi-level optimization. Experiments on four datasets show that our method brings 1 to 3 BLEU point improvement over baselines under the same latency. Our code is available at `https://github.com/P2F-research/simulNMT`.

## 1 Introduction

Neural machine translation (NMT) is an important task for the machine learning community and many advanced models have been designed (Sutskever et al., 2014; Bahdanau et al., 2014; Vaswani et al., 2017). In this work, we work on a more challenging task in NMT, simultaneous translation (also known as simultaneous interpretation), which is widely used in international conferences, summits and business. Different from standard NMT, simultaneous NMT has a stricter requirement for latency. We cannot wait to the end of a source sentence but have to start the translation right after reading the first few words. That is, the translator is required to provide instant translation based on a partial source sentence.

Simultaneous NMT is formulated as a prefix-to-prefix problem (Ma et al., 2019; 2020; Xiong et al., 2019), where a prefix refers to a subsequence starting from the beginning of the sentence to be translated. In simultaneous NMT, we face more uncertainty than conventional NMT, since the translation starts with a partial source sentence rather than the complete one. *Wait-k* inference (Ma et al., 2019) is a simple yet effective strategy in simultaneous NMT where the translation is $k$ words behind the source input. Rather than instant translation of each word, *wait-k* inference actually leverages $k$ more future words during inference phase. Obviously, a larger $k$ can bring more future information, and therefore results in better translation quality but at the cost of larger latency. Thus, when used in real-world applications, we should have a relatively small $k$ for simultaneous NMT.

While only small $k$ values are allowed in inference, we observe that *wait-m* training with $m > k$ will lead to better accuracy for *wait-k* inference. Figure 1 shows the results of training with *wait-m* but test with *wait-3* on IWSLT'14 English→German translation dataset. If training with $m = 3$, we will obtain a 22.79 BLEU score. If we set $m$ to larger values such as 7, 13 or 21 and test with *wait-3*, we can get better BLEU scores. That is, the model can benefit from the availability of more future information in training. This is consistent with the observation in (Ma et al., 2019).

The challenge is how much future information we should use in training. As shown in Figure 1, using more future information does not monotonically improve the translation accuracy of *wait-k* inference,

mainly because that more future information results in a larger mismatch between training and inference (i.e., $m - k$ more words are used in training than inference). Besides, due to the diversity of the natural language, intuitively, using different $m$'s for different sentences will lead to better performance. Even for the same sentence pair, the optimal $m$ for training might vary in different training stages. In this work, we propose an algorithm that can automatically determine how much future information to use in training for simultaneous NMT. Given a pre-defined $k$, we want to maximize the performance of *wait-k* inference. We have a set of $M$ training strategies *wait-m* with different waiting thresholds $m$ ($m \in \{1, 2, \cdots, M\}$). We introduce a controller such that given a training sample, the controller dynamically selects one of these training strategies so as to maximize the validation performance on *wait-k* inference. Which *wait-m* training strategy to select is based on the data itself and the network status of

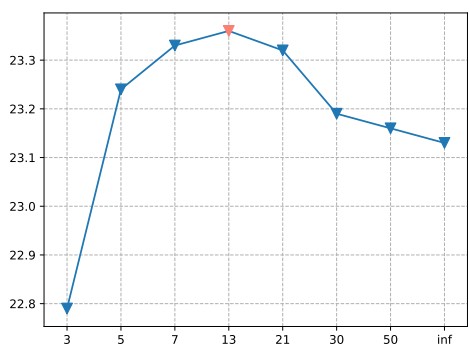

Figure 1: Preliminary exploration of IWSLT English-to-German simultaneous NMT. $x$-axis represents the waiting threshold $m$ during training and $y$-axis represents the BLEU scores testing with *wait-3*.

the current translation model. The controller and the translation model are jointly trained, and the learning process is formulated as a bi-level optimization problem (Sinha et al., 2018), where one optimization problem is nested within another.

Our contribution is summarized as follows:

(1) We propose a new method for simultaneous NMT, where a controller is introduced to adaptively determine how much future information to use for training. The controller and the translation model are jointly learned through bi-level optimization.

(2) Experiments on four datasets show that our method improves the *wait-k* baseline by 1 to 3 BLEU scores, and also consistently outperforms several heuristic baselines leveraging future information.

## 2 RELATED WORK

Previous work on simultaneous translation can be categorized by whether using a fixed decoding scheduler or an adaptive one. Fixed policies usually use pre-defined rules to determine when to read or to write a new token (Dalvi et al., 2018; Ma et al., 2019). *Wait-k* is the representative method for fixed scheduler (Ma et al., 2019), where the decoding is always $k$ words behind the source input. *Wait-k* achieves good results in terms of translation quality and controllable latency, and has been used in speech-related simultaneous translation (Zhang et al., 2019; Ren et al., 2020).

For methods that use adaptive schedulers, Cho & Esipova (2016) proposed wait-if-worse (WIW) and wait-if-diff (WID) methods which generate a new target word if its probability does not decrease (for WIW) or the generated word is unchanged (for WID) after reading a new source token. Grissom II et al. (2014) and Gu et al. (2017) used reinforcement learning to train the read/write controller, while Zheng et al. (2019a) obtained it in a supervised way. Alinejad et al. (2018) added a "predict" operator to the controller so that it can anticipate future source inputs. Zheng et al. (2019b) introduced a "delay" token into the target vocabulary indicating that the model should read a new word instead of generating a new one. Arivazhagan et al. (2019) proposed monotonic infinite lookback attention (MILk), which first used a hard attention model to determine when to read new source tokens, and then a soft attention model to perform translation. Ma et al. (2020) extended MILk into a multi-head version and proposed monotonic multihead attention (MMA) with two variants: MMA-IL (Infinite Lookback) which has higher translation quality by looking back at all available source tokens, and MMA-H(ard) which is more computational efficient by limiting the attention span.

Besides, Zheng et al. (2020a) extended *wait-k* to an adaptive strategy by training multiple *wait-m* models with different $m$'s and adaptively selecting a decoding strategy during inference. Zheng et al. (2020b) explored a new setting, where at each timestep, the translation model over generates the target words and corrects them in a timely fashion.

## 3 PROBLEM FORMULATION AND BACKGROUND

In this section, we first introduce the notations used in this work, followed by the formulation of *wait-k* strategy, and then we introduce our network architecture adapted from (Ma et al., 2019).

### 3.1 NOTATIONS AND FORMULATION

Let $\mathcal{X}$ and $\mathcal{Y}$ denote the source language domain and target language domain. For any $x \in \mathcal{X}$ and $y \in \mathcal{Y}$, let $x_i$ and $y_i$ denote the $i$-th token in $x$ and $y$ respectively. $L_x$ and $L_y$ are the numbers of tokens in $x$ and $y$. $x_{\leq t}$ represents a prefix of $x$, which is the subsequence $x_1, x_2, \cdots, x_t$, and similarly for $y_{\leq t}$. Let $D_{\mathrm{tr}}$ and $D_{\mathrm{va}}$ denote the training and validation sets, both of which are collections of bilingual sentence pairs.

The *wait-k* strategy (Ma et al., 2019) is defined as follows: given an input $x \in \mathcal{X}$, the generation of the translation $y$ is always $k$ tokens behind reading $x$. That is, at the $t$-th decoding step, we generate token $y_t$ based on $x_{\leq t+k-1}$ (more strictly, $x_{\leq \min\{t+k-1, L_x\}}$). Our goal is to obtain a model $f : \mathcal{X} \mapsto \mathcal{Y}$ with parameter $\theta$ that can achieve better results with *wait-k* inference.

### 3.2 MODEL ARCHITECTURE

Our model for simultaneous NMT is based on Transformer model (Vaswani et al., 2017). The model includes an encoder and a decoder, which are used for incrementally processing the source and target sentences respectively. Both the encoder and decoder are stacked of $L$ blocks. We mainly introduce the differences compared with the standard Transformer.

(1) *Incremental encoding*: Let $h_t^l$ denote the output of the $t$-th position from block $l$. For ease of reference, let $H_{\leq t}^l$ denote $\{h_1^l, h_2^l, \cdots, h_t^l\}$, and let $h_t^0$ denote the embeddings of the $t$-th token. An attention model $\mathtt{attn}(q, K, V)$, takes a query $q \in \mathbb{R}^d$, a set of keys $K$ and values $V$ as inputs. $K$ and $V$ are of equal size, $q \in \mathbb{R}^d$ where $d \in \mathbb{Z}_+$ is the dimension, $k_i \in \mathbb{R}^d$ and $v_i \in \mathbb{R}^d$ are the $i$-th key and value. $\mathtt{attn}$ is defined as follows:

$$\mathtt{attn}(q, K, V) = \sum_{i=1}^{|K|} \alpha_i W_v v_i, \ \alpha_i = \frac{\exp((W_q q)^\top (W_k k_i))}{Z}, \ Z = \sum_{i=1}^{|K|} \exp((W_q q)^\top (W_k k_i)), \tag{1}$$

where $W$'s are the parameters to be optimized. In the encoder side, $h_t^l$ is obtained in a unidirectional way: $h_t^l = \mathtt{attn}(h_t^{l-1}, H_{\leq t}^{l-1}, H_{\leq t}^{l-1})$. That is, the model can only attend to the previously generated hidden representations, and the computation complexity is $O(L_x^2)$. In comparison, (Ma et al., 2019) still leverages bidirectional attention, whose computation complexity is $O(L_x^3)$. We find that unidirectional attention is much more efficient than bidirectional attention without much accuracy drop (see Appendix D.1 for details).

(2) *Incremental decoding*: Since we use *wait-k* strategy, the decoding starts before reading all inputs. At the $t$-th decoding step, the decoder can only read $x_{\leq t+k-1}$. When $t \leq L_x - k$, the decoder greedily generates one token at each step, i.e., the token is $y_t = \arg\max_{w \in \mathcal{V}} P(w|y_{\leq t-1}; H_{\leq t+k-1}^L)$, where $\mathcal{V}$ is the vocabulary of the target language. When $t > L_x - k$, the model has read the full input sentence and can generate words using beam search (Ma et al., 2019).

## 4 OUR METHOD

We first introduce our algorithm in Section 4.1, and then we discuss its relationship with several other heuristic algorithms that leverage future information in Section 4.2.

### 4.1 ALGORITHM

Let $f(\cdots; \theta)$ denote a translation model parameterized by $\theta$, and let $\varphi$ denote the controller parameterized by $\omega$ to guide the training process of $f$. $f(\cdots; \theta^*(\omega))$ is the translation model obtained under the guidance of the controller $\varphi(\cdots; \omega)$, where $\theta^*(\omega)$ is the corresponding parameter. For each training data $(x, y)$, the controller $\varphi$ adaptively assigns a training task *wait-m*, where $m \in \{1, 2, \cdots, M\}$,

and $M \in \mathbb{Z}_+$ is a pre-defined hyperparameter. The input of $\varphi$ consists of two parts: (i) the information of the training data $(x, y)$; (ii) the network status of the translation model $f$. For ease of reference, denote these input features as $I_{x,y,f}$. We will discuss how to design $I_{x,y,f}$ in Section 5.1.

Let $\mathcal{M}_k(D_{\text{va}}; \theta^*(\omega))$ denote the validation metric, which is evaluated on the validation set $D_{\text{va}}$ with model $f(\cdots; \theta^*(\omega))$ and *wait-k* inference. We formulate the training process of $f$ and $\varphi$ as a bi-level optimization, where two optimization problems are nested together. In the inner-optimization, given a $\omega$, we want to obtain the model $f(\cdots, \theta^*(\omega))$ that can minimize the loss function $\ell$ on the training set $D_{\text{tr}}$ under the guidance of the controller $\varphi(\cdots, \omega)$. In the outer-optimization, given a translation model $\theta^*(\omega)$, we optimize $\omega$ to maximize the validation performance $\mathcal{M}_k$. The mathematical formulation is shown as follows:

$$\max_\omega \mathcal{M}_k(D_{\text{va}}; \theta^*(\omega));$$

$$\text{s.t. } \theta^*(\omega) = \arg\min_\theta \frac{1}{|D_{\text{tr}}|} \sum_{(x,y) \sim D_{\text{tr}}} \mathbb{E}_{m \sim \varphi(I_{x,y,f}; \omega)} \ell(x, y, m; \theta);$$

$$(2)$$

$$\text{where } \ell(x, y, m; \theta) = \sum_{(x,y)} \log P(y|x; \theta, m) = \sum_{(x,y)} \sum_{t=1}^{|y|} \log P(y_t | y_{\leq t-1}, x_{\leq t+m-1}).$$

We optimize Equation 2 in an alternative way, where we first optimize $\theta$ with a given $\omega$, and then update $\omega$ using the REINFORCE algorithm. We repeat the above process until convergence. Details are in Algorithm 1:

---

**Algorithm 1:** The optimization algorithm.

---

1   *Input*: Training episode $E$; internal update iterations $T$; learning rate $\eta_\theta$ of the translation model; learning rate $\eta_\omega$ of the controller; batch size $B$; initial parameters $\omega, \theta$;

2   **for** $e \leftarrow 1 : E$ **do**

3     Init a buffer to store states and actions: $\mathcal{B} = \{\}$;

4     **for** $t \leftarrow 1 : T$ **do**

5       Randomly sample a mini-batch of data $D_{e,t}$ from $D_{\text{tr}}$ with batch size $B$;

6       Assign a *wait-m* task to each data: $\tilde{D} = \{(x, y, m) | (x, y) \in D_{e,t}, m \sim \varphi(I_{x,y,f}; \omega)\}$, where the batch size is $B$, and $m$ is sampled from to the output distribution of $\varphi$;

7       Update the buffer: $\mathcal{B} \leftarrow \mathcal{B} \cup \{(I_{x,y,f}, m) | (x, y, m) \in \tilde{D}\}$;

8       Update the translation model: $\theta \leftarrow \theta - (\eta_\theta/B) \nabla_\theta \sum_{(x,y,m) \in \tilde{D}} \ell(x, y, m; \theta)$;

9     Calculate the validation performance as the reward: $R_e = \mathcal{M}_k(D_{\text{va}}; \theta_{e,T})$;

10    Update the controller: $\omega \leftarrow \omega + \eta_\omega R_e \sum_{(I,m) \in \mathcal{B}} \nabla_\omega \log P(\varphi(I; \omega) = m)$.

11   *Return* $\theta$.

---

Algorithm 1 consists of $E$ episodes (i.e., the outer loop), and each episode consists of $T$ update iterations (i.e., the inner loop). The inner loop (from step 4 to step 8) aims to optimize the $\theta$, where we can update the parameter with any gradient based optimizer like momentum SGD, Adam (Kingma & Ba, 2015), etc. The outer loop (from step 2 to step 10) aims to optimize $\omega$. $\varphi(I_{x,y,f}; \omega)$ can be regarded as a policy network, where the state is $I_{x,y,f}$, the action is the choice of the task *wait-m*, $m \in \{1, 2, \cdots, M\}$, and the reward is the validation performance $R_e$ (step 9). At the end of each episode, we update $\omega$ using REINFORCE algorithm (step 10).

## 4.2 DISCUSSION

To use more information and to obtain better *wait-k* inference, there are several heuristic methods:

(1) *Wait-$k^*$*: We train $M$ translation models using the *wait-m* strategy, $m \in \{1, 2, \cdots, M\}$, select the best waiting threshold $k^*$ according to the validation performance, and use the corresponding model for *wait-k* inference.

(2) Random sampling (briefly, Random): For each training data, randomly choose $m$ from $\{1, 2, \cdots, M\}$ with equal probability and using *wait-m* training.

(3) Curriculum learning (briefly, CL): We gradually decrease $m$ from $M$ to the threshold $k$ we will use in the inference (see Appendix A.2 for mathematical definition).

However, a common drawback of them is that they cannot dynamically adjust $k$ according to the sentence representation and the model status. In *wait-$k^*$*, during training, each sentence pair is treated with the same $k^*$. For Random and CL, the selection of waiting threshold $m$ is predefined and not adjusted according to the training. We overcome this difficulty by introducing a controller, which is trained via reinforcement learning to maximize the validation performance, and is able to adaptively determine how much exploration the model requires and how long we should use a specific *wait-$m$* strategy. Therefore, our method is expected to outperform *wait-$k^*$*, Random and CL.

We also extend our method to an adaptive version by combining with Zheng et al. (2020a). With a set of pre-trained *wait-$m$* models with different $m$ values, Zheng et al. (2020a) adaptively selects the waiting threshold during inference. Therefore, we can combine Zheng et al. (2020a) with our method, where the *wait-$m$* models are obtained through our strategy.

## 5 EXPERIMENTS

We work on the text-to-text simultaneous NMT in this paper. Let us briefly denote English, German, Vietnamese and Chinese as En, De, Vi and Zh respectively. We conduct experiments on three small-scale datasets: IWSLT'14 En→De, IWSLT'15 En→Vi and IWSLT'17 En→Zh, and a large-scale dataset: WMT'15 En→De translation.

### 5.1 SETTINGS

*Datasets*: For IWSLT'14 En→De, following (Edunov et al., 2018), we split $7k$ sentences from the training corpus for validation, and the test set is the concatenation of *tst2010, tst2011, tst2012, dev2010* and *dev2012*. For IWSLT'15 En→Vi, following (Ma et al., 2020), we use *tst2012* as the validation set and *tst2013* as the test set. For IWSLT'17 En→Zh, we concatenate *tst2013*, *tst2014* and *tst2015* as the validation set and use *tst2017* as the test set. For WMT'15 En→De, following (Ma et al., 2019; Arivazhagan et al., 2019), we use *newstest2013* as the validation set and use *newstest2015* as the test set. More details about datasets can be found at Appendix B.1.

*Models*: The translation model $f$ is based on Transformer. For IWSLT En→Zh and En→Vi, we use the transformer small model, where the embedding dimension, feed-forward layer dimension, number of layers are 512, 1024 and 6 respectively. For IWSLT En→De, we use the same architecture but change the embedding dimension into 256. For WMT'15 En→De, we use the transformer big setting, where the above three numbers are 1024, 4096 and 6 respectively. The controller $\varphi$ for each task is a multilayer perceptron (MLP) with one hidden layer and the `tanh` activation function. The size of the hidden layer is 256.

*Input features of $\varphi$*: The input $I_{x,y,f}$ is a 7-dimension vector with the following features: (1) the ratios between the lengths of the source/target sentences to the average source/target sentence lengths in all training data (2 dimensions), i.e., $L_x/(\sum_{x' \in \mathcal{X}} L_{x'}/|\mathcal{X}|)$ and $L_y/(\sum_{y' \in \mathcal{Y}} L_{y'}/|\mathcal{Y}|)$; (2) the training loss over data $(x, y)$ evaluated by *wait-$k$*; (3) the average of historical training losses; (4) the validation loss of the previous epoch; (5) the average of historical validation loss; (6) the ratio of current training step to the total training iteration.

*Choice of $\mathcal{M}_k$*: The validation performance $\mathcal{M}$ is the negative validation loss with *wait-$k$* strategy. To stabilize the training, we minus a baseline to the $R_e$ in step 9 of algorithm 1, which is the validation loss of the previous episode, i.e., $R_{e-1}$. That is, the reward signal at episode $e$ is $R_e - R_{e-1}$. $R_0$ is the negative validation loss of the randomly initialized model.

*Baselines*: We implement the *wait-$k^*$*, Random and CL baselines discussed in Section 4.2. We also compare our algorithm with several adaptive methods, including Wait-if-Worse (WIW), Wait-if-Diff (WID), MILk, MMA-IL and MMA-H (refer to Section 2 for a brief introduction). As discussed in Section 4.2, we also compare and combine our method with Zheng et al. (2020a). We leave the training details of all algorithms (optimizer, hyperparameter selection, etc) in Appendix B.2, and the implementation details of baseline algorithms in Appendix B.3.

*Evaluation*: We use BLEU to measure the translation quality, and use Average Proportion (AP) and Average Lagging (AL) to evaluate translation delay. AP measures the average proportion of source symbols required for translation, and and AL measures the average number of delayed words, which is complementary to AP (see Appendix A.1 for details). Following the common practice (Ma et al., 2019; 2020), we show the BLEU-AP and BLEU-AL curves to demonstrate the tradeoff between quality and latency. For IWSLT'14 En→De and IWSLT'15 En→Vi, we use `multi-bleu.perl` to evaluate the BLEU scores; for IWSLT'17 En→Zh and WMT'15 En→De, we use `sacreBLEU` to evaluate the detokenized BLEU scores.

## 5.2 RESULTS

We first compare our method with the baseline methods on IWSLT datasets. The BLEU-latency curves are shown in Figure 2, and the BLEU scores of En→Vi under different *wait-k* inference are reported in Table 1. The BLEU scores of all language pairs are left in Appendix C.

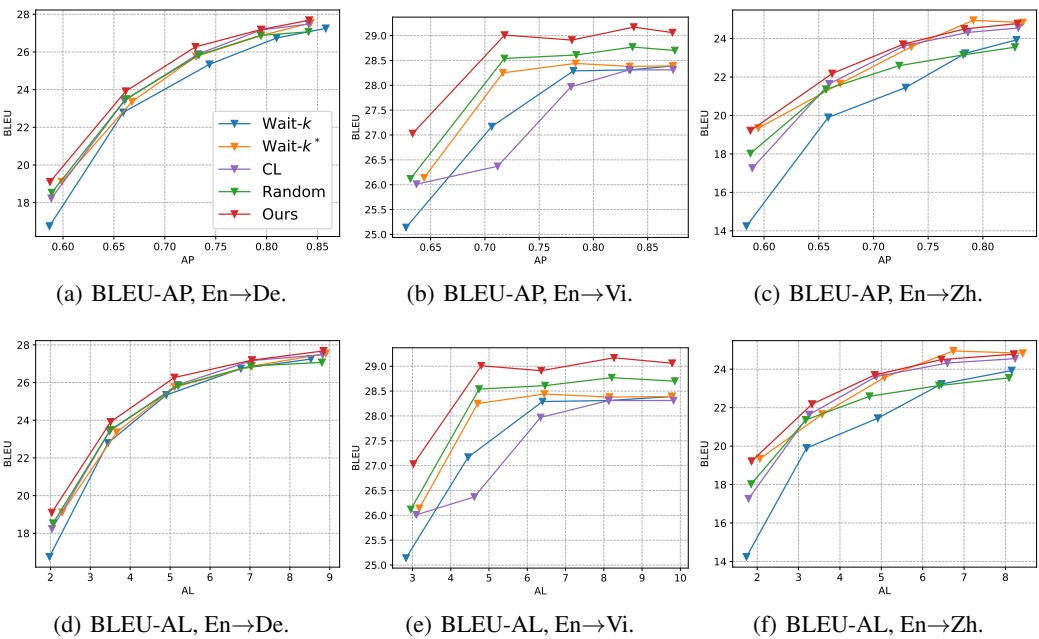

Figure 2: Translation quality against latency metrics (AP and AL) on IWSLT'14 En→De, IWSLT'15 En→Vi and IWSLT'17 En→Zh tasks .

| Test $k$ | *wait-k* | *wait-k*\*/ $k^*$ | CL | Random | Ours |
|---|---|---|---|---|---|
| 1 | 25.14 | 26.14 / 5 | 26.01 | 26.12 | 27.03 |
| 3 | 27.17 | 28.25 / 5 | 26.37 | 28.54 | 29.01 |
| 5 | 28.29 | 28.44 / 9 | 27.97 | 28.61 | 28.91 |
| 7 | 28.31 | 28.38 / 13 | 28.31 | 28.77 | 29.17 |
| 9 | 28.39 | 28.39 / 9 | 28.31 | 28.70 | 29.06 |

Table 1: BLEU scores on IWSLT En→Vi simultaneous NMT tasks.

We have the following observations:

(1) Generally, our method consistently performed the best across different translation tasks in terms of both translation quality and controllable latency. As shown in Table 1, our method achieves the highest BLEU scores among all baselines. We perform significance test on En→Vi and find that our method significantly outperforms *wait-k* ($p < 0.01$ for *wait*-1,3 and 9; $p < 0.05$ for *wait*-5 and 7). In Figure 2, the curve for our method (i.e., the red one) is on the top in most cases, which indicates that given specific latency (e.g., AP or AL), we can achieve the best translation quality.

(2) Baselines like *wait-k\**, Random and CL can also outperform the vanilla *wait-k*, which demonstrates the effectiveness of leveraging future information. However, the improvements are not consistent, and it is hard to tell which baseline is better. On En→De, the performance of the three baselines is similar and CL slightly outperforms the other two. On En→Zh, *wait-k\** performs the best followed by CL which performs well at higher latency. In comparison, the improvement brought by our method is much more consistent.

(3) The improvement brought by our method is more significant with smaller $k$'s than that with bigger $k$'s. We observe that all baselines perform well with bigger $k$, where more information is available during inference. That is, the advantages of leveraging future information are less significant. We provide analysis on the training and validation loss in Appendix **??**, which shows that leveraging future information can improve generalization ability.

The results of WMT'15 En→De, whose training corpus is larger, are shown in Figure 3. The heuristic methods (*wait-k\**, Random, CL) do not bring much improvement compared to *wait-k*. Our method consistently outperforms all baselines, which demonstrates the effectiveness of our method on large datasets. We further evaluate them on WMT'14 and WMT'16 test sets and obtain similar conclusions (see Figure 7 in Appendix C for details).

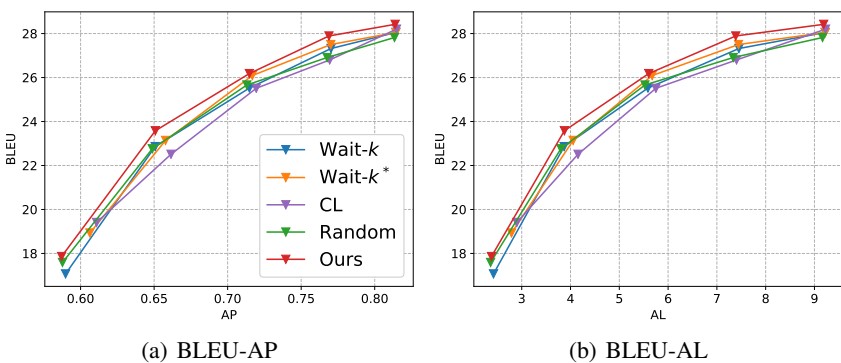

(a) BLEU-AP       (b) BLEU-AL

Figure 3: Translation quality against latency metrics (AP and AL) on WMT'15 En→De.

We further compare our method with WIW, WID, MILk, MMA-IL and MMA-H on IWSLT En→Vi. The BLEU-AL curves are shown in Figure 4(a) and the BLEU-AP curves are in Appendix C. When AL $\geq 5.0$, our method outperforms all baseline models, and when AL $< 5.0$, our method performs slightly worse than MMA-IL and MMA-H, since ours does not take reducing latency into consideration explicitly but focuses on improving performance under given waiting thresholds.

To verify the effectiveness of the adaptive extension of our method, we compare and combine our method with Zheng et al. (2020a) , where the *wait-m* models are obtained through the vanilla *wait-k* (denoted by "Zheng et al.") and our strategy respectively (denoted by "+Ours". The BLEU-AL curves are shown in Figure 4(b), and the BLEU-AP curves are in Appendix C. We can see that: (1) our method catches up with (Zheng et al., 2020a), which is built upon 10 models in total (*wait-1* to *wait-10*); (2) after combing our approach with Zheng et al. (2020a), the performance can be further improved, which shows that our method is complementary to adaptive inference strategies like Zheng et al. (2020a).

## 5.3 COMPUTATIONAL OVERHEAD

To evaluate the additional computational overhead brought by our method, we compare the training speed of *wait-3* (measured by batch per second) of standard *wait-k* and our method. Results on IWSLT datasets are summarized in Table 2. Our method requires $20\% \sim 30\%$ additional training time, which is acceptable considering the improvements of performance. The main overhead is from computing the training loss by *wait-k*. To verify that, we record the training speed of our method without the second and third input features of $\varphi$, which are the training loss over data $(x, y)$ evaluated by *wait-k*; and the average of historical training losses. Without them, the training speed of our method is similar to *wait-k* (i.e., the "Ours w/o feature (2,3)").

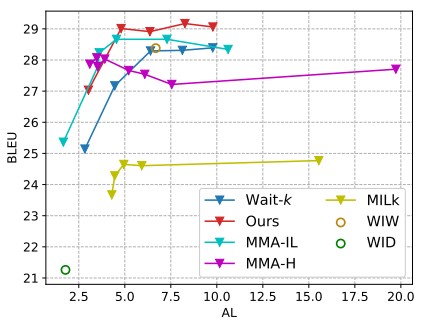

(a) Comparison of our method, WIW, WID, MILk, MMA-IL and MMA-H.

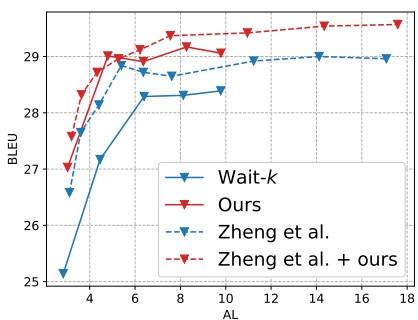

(b) Comparison and combination of our method and Zheng et al. (2020a).

Figure 4: BLEU-AL comparison between our method and baselines on En→Vi.

| Task | wait-$k$ | Ours | Ours w/o feature (2,3) |
|------|---------|------|------------------------|
| En→De | 5.3 | 4.0 (-23%) | 5.2 (-2%) |
| En→Vi | 1.5 | 1.1 (-27%) | 1.4 (-7%) |
| En→Zh | 2.5 | 1.8 (-28%) | 2.4 (-4%) |

Table 2: Comparison of training speed (batch / sec) between wait-$k$ and our methods.

## 5.4 ANALYSIS

**(I) Strategy analysis**: In Figure 5, we visualize the distribution of wait-$m$ training strategies obtained by our algorithm for wait-3 and wait-9 inference on IWSLT En→Zh task. We show the frequency of each wait-$m$ strategy sampled by the controller $\varphi$ at the 0th, 1st, 5th, 10th and 40th episode.

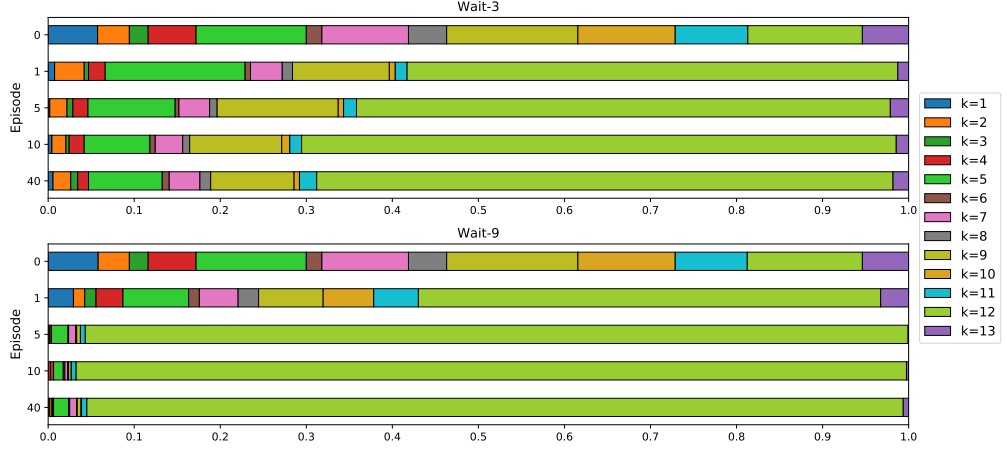

Figure 5: An illustration of the strategies for wait-3 and wait-9 on En→Zh dataset.

We observed that the controller uniformly samples different $m$'s at first, and then the strategies converge within 10 episodes. After convergence, the controller mainly samples several specific actions, i.e., $m = 5, 9, 12$ for wait-3, and $m = 5, 12$ for wait-9. The action that both controllers prefer most is $m = 12$, which is close to the wait-$k^*$ strategy ($k^* = 11$) for both wait-3 and wait-9. Generally, these two strategies assign most of the sampling frequency to large $m$, which again shows the importance of using future information. However, it is worth noting that the controller also samples smaller $m$, which means that the past information is also utilized. For example, the controller for wait-9 still samples $m = 5$ with a probability about 0.02. Our conjecture is that the use of past

information helps mitigate the mismatch between training and inference. If the model is always trained with future information, this mismatch will be large.

**(II) Action space selection**: In previous experiments, both future information and past information are leveraged. We want to study the effect of using past information or future information only. For any *wait-k*, we build two other action spaces for $\varphi$: $\mathcal{K}_p(k) = \{1, 2, \cdots, k\}$; $\mathcal{K}_f(k) = \{k, k+1, \cdots, K\}$. We evaluate *wait*-3, 5 on IWSLT'14 En→De with the above two action spaces.

The results are reported in Table 3. We observe that our method with full action space significantly outperforms that with $\mathcal{K}_p(k)$ and slightly outperforms that using $\mathcal{K}_f(k)$. This shows that leveraging both kinds of information can help improve the performance.

|  | $k = 3$ | | | $k = 5$ | | |
|---|---|---|---|---|---|---|
|  | BLEU | AP | AL | BLEU | AP | AL |
| Full action space | 23.91 | 0.650 | 3.252 | 26.27 | 0.723 | 4.887 |
| $\mathcal{K}_f(k)$ | 23.70 | 0.655 | 3.386 | 26.04 | 0.730 | 5.134 |
| $\mathcal{K}_p(k)$ | 22.80 | 0.645 | 3.078 | 25.58 | 0.726 | 4.979 |

Table 3: Ablation study for feature selection on IWSLT'14 En→De dataset.

**(III) Feature selection**: To emphasize the importance of the selected features in Section 5.1, we provide four groups of ablation study, where in each group some specific features are excluded: (i) source and target sentence lengths; (ii) current training loss and average historical training loss; (iii) current validation loss and average historical validation loss; (iv) training step. We work on IWSLT'14 En→De task and study the effect to *wait*-3, 5, 7,

The results are shown in Table 4. We report the BLEU scores only, since the latency metrics (AP and AL) are not significantly influenced. Removing any feature causes performance drop, indicating that they all contribute to the decision making. Specifically, network status information including validation performance (feature iii) and training stage (feature iv) is more important than input data information including sequence length (feature i) and data difficulty (feature ii).

|  | $k = 3$ | $k = 5$ | $k = 7$ |
|---|---|---|---|
| Ours | 23.91 | 26.27 | 27.19 |
| - (i) | 23.67 (-1.00%, rank 3) | 26.03 (-0.91%, rank 3) | 26.92 (-0.99%, rank 4) |
| - (ii) | 23.70 (-0.88%, rank 4) | 26.04 (-0.88%, rank 4) | 26.91 (-1.03%, rank 3) |
| - (iii) | 23.57 (-1.42%, rank 1) | 25.92 (-1.33%, rank 2) | 26.72 (-1.73%, rank 1) |
| - (iv) | 23.65 (-1.09%, rank 2) | 25.63 (-2.44%, rank 1) | 26.86 (-1.21%, rank 2) |

Table 4: Ablation study for feature selection on IWSLT'14 En→De dataset.

## 6 CONCLUSION AND FUTURE WORK

In this work, we proposed a new approach for simultaneous NMT. Motivated by the fact that *wait-k* benefits from future information, we introduced a controller, which adaptively assigns a training task *wait-m* to each input. A bi-level optimization method is leveraged to jointly obtain the translation model and the controller. Experiments on four translation tasks demonstrate the effectiveness of our approach. For future work, first, we will enhance the objective function in Eqn.(2) beyond using translation quality only and explicitly introduce the latency constraint. Second, we will combine our method with the adaptive decoding methods (Arivazhagan et al., 2019; Ma et al., 2020). Third, we will apply the idea in this work to more applications like action prediction (Kong et al., 2020; Cai et al., 2019), weather forecasting, game AI (Li et al., 2020; Vinyals et al., 2019), etc.

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

## A    MATHEMATICAL DEFINITIONS

### A.1    LATENCY METRICS DEFINITIONS

Given the input sentence $x$ and the output sentence $y$, let $L_x$ and $L_y$ denote the length of $x$ and $y$ respectively. Define a function $g(t)$ of decoding step $t$, which denotes the number of source

tokens processed by the encoder when deciding the target token $y_t$. For *wait-k* strategy, $g(t) = \min\{t + k - 1, L_x\}$. The definition of Average Proportion (AP) and Average Lagging (AL) are listed in Equation 3 and Equation 4.

$$\text{AP}_g(x, y) = \frac{1}{|x||y|} \sum_{t=1}^{|y|} g(t); \tag{3}$$

$$\text{AL}_g(x, y) = \frac{1}{\tau_g(|x|)} \sum_{t=1}^{\tau_g(|x|)} \left( g(t) - \frac{t - 1}{|y|/|x|} \right), \tag{4}$$

$$\text{where} \quad \tau_g(|x|) = \min\{t | g(t) = |x|\}.$$

We use the scripts provided by Ma et al. (2019) to calculate AP and AL scores.

### A.2 MATHEMATICAL FORMULATION OF CURRICULUM LEARNING

In the curriculum learning (briefly, CL) baseline, we gradually decrease $m$ from $K$ to the threshold $k$ which will used in the test setting. The mathematical formulations are shown as follows:

$$m = M - \lfloor \frac{t - 1}{T} (M - k + 1) \rfloor \tag{5}$$

where $T$ denotes the total update number, $t$ denotes the current update number ($t = 1, 2, ..., T$).

## B    MORE DETAILED SETTINGS ABOUT EXPERIMENTS

### B.1    DETAILED INTRODUCTION OF THE DATASETS

For IWSLT'14 En→De, following Edunov et al. (2018), we lowercase all words, tokenize them and apply BPE with $10k$ merge operations jointly to the source and target sequences. We split $7k$ sentences from the training corpus for validation and the remaining $160k$ sequences are left as the training set. The test set is the concatenation of *tst2010, tst2011, tst2012, dev2010* and *dev2012*, which consists of 6750 sentences.

For IWSLT'15 En→Vi, following Ma et al. (2020), we tokenize the data and replace words with frequency less than 5 by <unk>[1]. We use *tst2012* as the validation set and *tst2013* as the test set. The training, validation and test sets contains $133k$, 1268 and 1553 sentences respectively.

For IWSLT'17 En→Zh, we tokenize the data and apply BPE with $10k$ merge operations independently to the source and target sequences[2]. We use the concatenation of *tst2013*, *tst2014* and *tst2015* as the validation set and use *tst2017* as the test set. The training, validation and test sets contains $235k$, 3874 and 1459 sentences respectively. For WMT'15 En↔De, we follow the setting in Ma et al. (2019); Arivazhagan et al. (2019). We tokenize the data, apply BPE with $32k$ merge operations jointly to the source and target sentences, and get a training corpus with $4.5M$ sentences. We use *newstest2013* as the validation set and use *newstest2015* as the test set.

### B.2    DETAILED TRAINING STRATEGY

For the translation model, we use Adam (Kingma & Ba, 2015) optimizer with initial learning rate $5 \times 10^{-4}$ and `inverse_sqrt` scheduler (see Section 5.3 of (Vaswani et al., 2017) for details). The batch size and the number of GPUs of IWSLT En→De, En→Vi, En→Zh and WMT'15 En→De are $4096 \times 1\text{GPU}$, $16000 \times 1\text{GPU}$, $4000 \times 1\text{GPU}$ and $3584 \times 8 \times 16\text{GPU}$ respectively. For IWSLT tasks, the learning rate $\eta$ is grid searched from $\{5 \times 10^{-4}, 5 \times 10^{-5}, 5 \times 10^{-6}, 5 \times 10^{-7}\}$ with vanilla SGD optimizer, and the internal update iteration $T$ is grid searched from $\{\frac{1}{2}t, t, 2t\}$, where $t$ is the number of updates in an epoch of the translation model training. For WMT'15 En→De, due to resource limitation, we do not train the translation model from scratch. The translation model is warm started from pretrained *wait-k* model, the learning rate is set as $5 \times 10^{-5}$, and the internal update iteration $T$ is 16.

---

[1]The data is downloaded from `https://nlp.stanford.edu/projects/nmt/`, which has been tokenized already.

[2]The Chinese sentences are tokenized using Jieba ( `https://github.com/fxsjy/jieba` ).

### B.3 DETAILED BASELINE IMPLEMENTATION

In this section, we introduce how we reproduce baseline models and get the results on En→Vi task.

For MILk (Arivazhagan et al., 2019) and MMA (Ma et al., 2020), we do not reproduce the results, but directly use the results reported in Ma et al. (2020). Note that the results for MILk is reproduced by Ma et al. (2020).

For WIW and WID (Cho & Esipova, 2016), we pre-train a standard translation model, using the exact same architecture and training hyperparameters as our method and in (Ma et al., 2020). For fair comparison, we adopt bi-directional attention.

For Zheng et al. (2020a), we use the pre-trained *wait-k* model and the model obtained through our results ($k = 1, 2, 3, ..., 10$). We use only single model rather than ensemble. As in Zheng et al. (2020a), the thresholds of different $k$ values are obtained in this way: $\rho_i = \rho_1 - (i-1) * (\rho_1 - \rho_{10})/9$, where we test with $\rho_1 \in \{0.2, 0.4, 0.6, 0.8\}$, $\rho_{10} = 0$; and $\rho_1 = 1$, $\rho_{10} \in \{0, 0.2, 0.4, 0.6, 0.8\}$.

## C SUPPLEMENTAL RESULTS

In this section, we report the specific BLEU scores and some additional results. The BLEU scores for IWSLT tasks and WMT'15 En→De task are reported in Table 5 and Table 6 respectively. We further evaluate the baselines and our methods on WMT'14 and WMT'18 test sets, and report the BLEU-latency curves in Figure 7. The BLEU-AL curves of our methods and baselines on IWSLT'15 En→Vi are reported in Figure 6(a), the BLEU-AL curves of our method and Zheng et al. (2020a) are in Figure 6(b).

| Task | *wait-k* | *wait-k\**/ best $k^*$ | CL | Random | Ours |
|------|------|------|------|------|------|
| En→De ($k = 1$) | 16.75 | 19.11 / 9 | 18.23 | 18.53 | 19.10 |
| En→De ($k = 3$) | 22.79 | 23.36 / 13 | 23.41 | 23.50 | 23.91 |
| En→De ($k = 5$) | 25.34 | 25.76 / 11 | 25.88 | 25.84 | 26.27 |
| En→De ($k = 7$) | 26.74 | 26.87 / 9 | 26.85 | 26.88 | 27.19 |
| En→De ($k = 9$) | 27.25 | 27.54 / 11 | 27.48 | 27.07 | 27.68 |
| En→Vi ($k = 1$) | 25.14 | 26.14 / 5 | 26.01 | 26.12 | 27.03 |
| En→Vi ($k = 3$) | 27.17 | 28.25 / 5 | 26.37 | 28.54 | 29.01 |
| En→Vi ($k = 5$) | 28.29 | 28.44 / 9 | 27.97 | 28.61 | 28.91 |
| En→Vi ($k = 7$) | 28.31 | 28.38 / 13 | 28.31 | 28.77 | 29.17 |
| En→Vi ($k = 9$) | 28.39 | 28.39 / 9 | 28.31 | 28.70 | 29.06 |
| En→Zh ($k = 1$) | 14.24 | 19.34 / 9 | 17.26 | 18.02 | 19.21 |
| En→Zh ($k = 3$) | 19.90 | 21.66 / 11 | 21.64 | 21.36 | 22.18 |
| En→Zh ($k = 5$) | 21.45 | 23.57 / 11 | 23.62 | 22.59 | 23.70 |
| En→Zh ($k = 7$) | 23.23 | 24.95 / 11 | 24.32 | 23.15 | 24.35 |
| En→Zh ($k = 9$) | 23.93 | 24.83 / 11 | 24.55 | 23.55 | 24.78 |

Table 5: BLEU scores on IWSLT simultaneous NMT tasks.

| $k$ | *wait-k* | *wait-k\**/ best $k^*$ | CL | Random | Ours |
|-----|------|------|------|------|------|
| 1 | 17.07 | 19.83 / 9 | 19.41 | 17.59 | 18.14 |
| 3 | 22.86 | 23.14 / 7 | 22.51 | 22.76 | 23.58 |
| 5 | 25.52 | 26.09 / 7 | 25.51 | 25.66 | 26.18 |
| 7 | 27.32 | 27.50 / 9 | 26.80 | 26.91 | 27.89 |
| 9 | 28.05 | 28.05 / 9 | 28.20 | 27.82 | 28.42 |

Table 6: BLEU scores on WMT'15 En→De dataset.

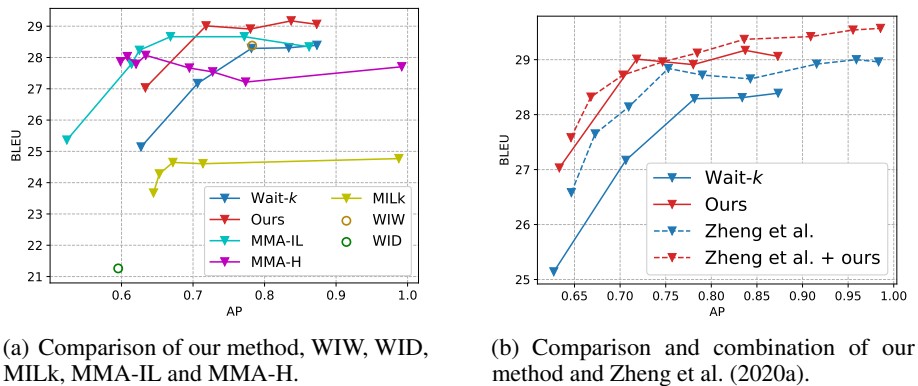

(a) Comparison of our method, WIW, WID, MILk, MMA-IL and MMA-H.

(b) Comparison and combination of our method and Zheng et al. (2020a).

Figure 6: BLEU-AP comparison between our method and baselines on En→Vi.

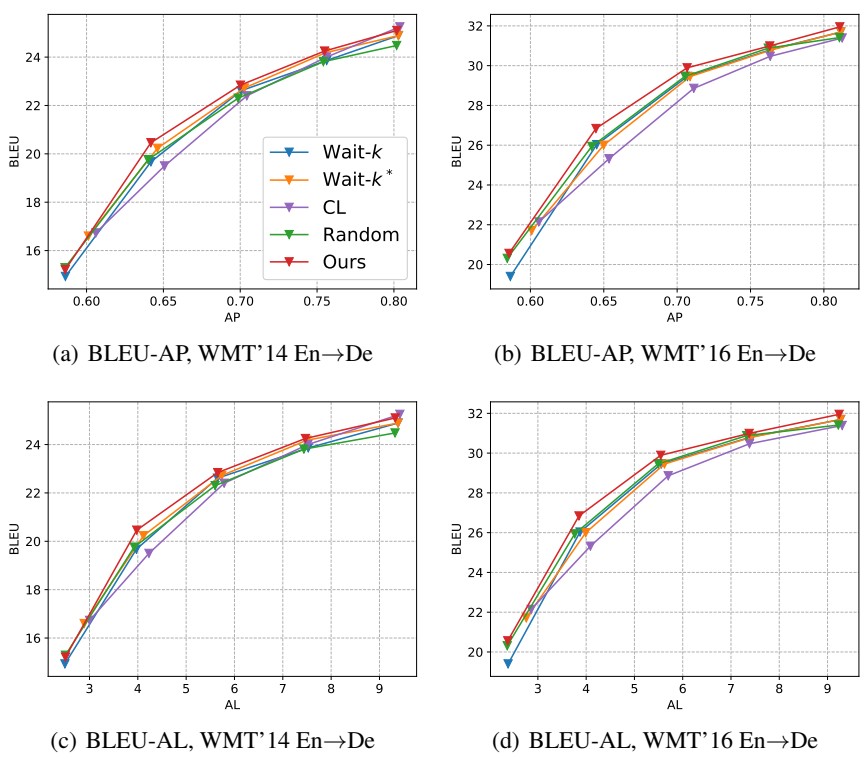

(a) BLEU-AP, WMT'14 En→De

(b) BLEU-AP, WMT'16 En→De

(c) BLEU-AL, WMT'14 En→De

(d) BLEU-AL, WMT'16 En→De

Figure 7: Translation quality against latency metrics (AP and AL) on WMT'14 and 16 English→German test sets.

# D    ADDITIONAL ABLATIONS AND ANALYSIS

## D.1    MODEL ARCHITECTURE SELECTION

As mentioned in Section 3 of the main content,we adopt unidirectional attention instead of bidirectional attention in the encoder side. We compare the performance the *wait-k* model with two attention types on IWSLT'14 En→De dataset, and the results are in Figure 8(a) and Figure 8(b). On IWSLT'14, we observe that the performance of *wait-k* with unidirectional attention slightly drops than that with bidirectional attention. On WMT'15 En→De dataset, our implementation of *wait-k* with unidirectional attention is slightly better than that of bidirectional attention reported in Ma et al. (2019). However, the computational cost of bidirectional attention is much larger than unidirectional attention. For example, the inference speed of unidirectional *wait-9* model is $57.39$ sentences / second, while the inference speed of bidirectional attention is $6.48$ sentences / second.

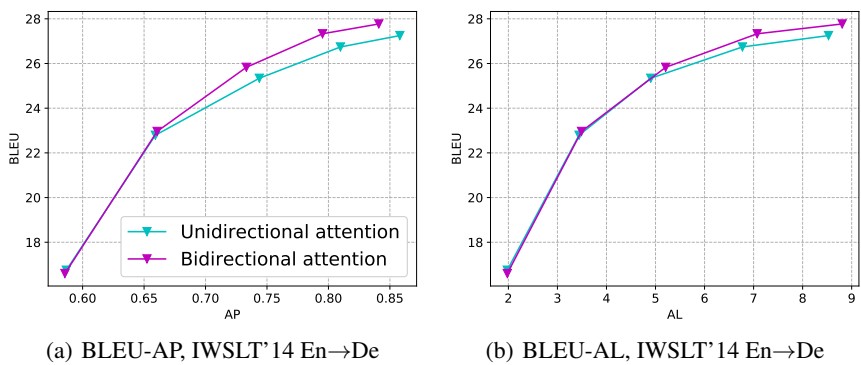

(a) BLEU-AP, IWSLT'14 En→De          (b) BLEU-AL, IWSLT'14 En→De

Figure 8: Ablation study of different model architectures on IWSLT'14 En→De dataset and WMT'15 En→De dataset.

## D.2    CASE STUDY

To analyze the effect of using future information, we present two translation examples for En→Zh *wait-3* translation in Table 7 and Table 8. We observe that all methods tend to anticipate when the future information is lacking (Table 7). *Wait-3* makes more mistake (Table 7) and even makes wrong anticipation where there is no need to anticipate (Table 8), while *wait-k\** and Ours anticipate more appropriately (Table 7). However, as in Table 8, *wait-k\** sometimes generates repeated information, therefore increasing the overall latency. This might be resulted from the gap between training and testing, as *wait-k\** is trained to produce higher latency. Our method can leverage the advantages of both methods, and produces translations with the best quality.

## D.3    MORE HEURISTIC BASELINES

In this section, we conduct ablation studies on two more heuristic baselines to demonstrate the effectiveness of our method.

(1) **Randomly selecting $k$ in a window around $k^*$**: We implement another baseline which is a combination of *wait-k\** and Random. After obtaining $k^*$, instead of sampling *wait-m* on all possible $\{1, 2, \cdots, M\}$, we sample on a smaller region around $k^*$. We conduct experiments on IWSLT En→De and the results are in Figure 9. We can see that this variant achieves similar results with *wait-k\**, but is not as good as our method. This shows the importance of using an adaptive controller to guide the training.

(2) **A variant of CL**: We implement self-paced learning (SPL) for *wait-k*, a CL method based on loss function: within each minibatch, we remove the $\tau\%$ sentences with the largest loss, where $\tau$ is gradually decreased from $40$ to $0$. On IWSLT En→De, when $k = 3$, the BLEU/AL/AP for SPL are $22.87/0.66/3.47$, which are worse than conventional CL ($23.41/0.66/3.48$).

| 1 | 2 | 3 | 4 | 5 | 6 | 7 | 8 | 9 | 10 | 11 | 12 | 13 | 14 | 15 | 16 |
|---|---|---|---|---|---|---|---|---|---|---|---|---|---|---|---|
| I | was | born | with | epi@@ | le@@ | p@@ | sy | | and | an | intellectual | disability | . | | |

| *wait-3* | 我 | 出生 | 在 | 一个 | 充满 | 癫@@ | 痫 | 的 | 知识@@ | 产@@ | 障碍 | 的 | 国家 | 。 |
|---|---|---|---|---|---|---|---|---|---|---|---|---|---|---|
| | I | was born | in | a | full of | | epilepsy | - | | Not a word | | - | country | . |

I was born in a country full of epilepsy Not a word.

| *Wait-k\** | 我 | 出生 | 的 | 时候 | , | 我 | 患有 | 癫@@ | 痫 | 和 | 智力 | 障碍 | 。 |
|---|---|---|---|---|---|---|---|---|---|---|---|---|---|
| | I | was born | - | when | , | I | suffered from | | epilepsy | and | intellectual | disability | . |

When I was born, I suffered from epilepsy and intellectual disability.

| *Ours* | 我 | 出生 | 时 | , | 伴随 | 着 | 癫@@ | 痫 | 和 | 智力 | 障碍 | 。 |
|---|---|---|---|---|---|---|---|---|---|---|---|---|
| | I | was born | when | , | with | - | | epilepsy | and | intellectual | disability | . |

When I was born, I was accompanied by epilepsy and intellectual disability.

Table 7: Example 1 for En→Zh *wait*-3 translation. In this example and the next example, different colors represent different meanings. Specifically, green and red represents information that does not exist in the source sentence (i.e., anticipated by the model), where green represents information that is consistent with the input sentence (i.e. correctly anticipated), and red represents information that is inconsistent with the input sentence (i.e., wrongly anticipated).

At step 5, *Wait*-3 anticipates "在一个" (in a), while *wait-k\** and Ours anticipates "的时候" (when) and "时" (when) respectively. The anticipation generated by *wait-k\** and Ours are more appropriate within the context, while *wait*-3 makes mistakes.

| 1 | 2 | 3 | 4 | 5 | 6 | 7 | 8 | 9 | 10 | 11 | 12 | 13 | 14 | 15 | 16 | 17 | 18 | 19 |
|---|---|---|---|---|---|---|---|---|---|---|---|---|---|---|---|---|---|---|
| And | I | opened | up | the | website | , | and | there | was | my | face | staring | right | back | at | me | . | |

| *wait-3* | 我 | 打开 | 了 | 网站 | , | 我 | 发现 | 了 | 我 | 的 | 脸 | 。 |
|---|---|---|---|---|---|---|---|---|---|---|---|---|
| | I | opened | - | the website | , | I | found | - | I | *POS* | face | . |

I opened the website and I found my face.

| *Wait-k\** | 我 | 打开 | 了 | 网站 | , | 打开 | 了 | 网站 | , | 我 | 的 | 脸 | 就 | 在 | 看着 | 我 | 。 |
|---|---|---|---|---|---|---|---|---|---|---|---|---|---|---|---|---|---|
| | I | opened | - | the website | , | opened | the | website | , | I | *POS* | face | *PROG* | *PROG* | looking | at me | . |

I opened the website, opened the website, and my face was looking at me.

| *Ours* | 我 | 打开 | 了 | 网站 | , | 然后 | 就 | 有 | 了 | 我 | 的 | 脸 | 盯 | 着 | 我 | 。 |
|---|---|---|---|---|---|---|---|---|---|---|---|---|---|---|---|---|
| | I | opened | - | the website | , | then | - | there was | - | I | *POS* | face | stare | *PROG* | me | . |

I opened the website, and then there was my face staring at me.

Table 8: Example 2 for En→Zh *wait*-3 translation, where *POS* indicates possessive forms, and *PROG* indicates progressive tense. In this example, there is no need to anticipate. However, *wait*-3 still anticipates "发现" (found) and makes a mistake. *Wait-k\** makes a mistake by repeating "打开了网站" (opened the website). Ours generates the best translation.

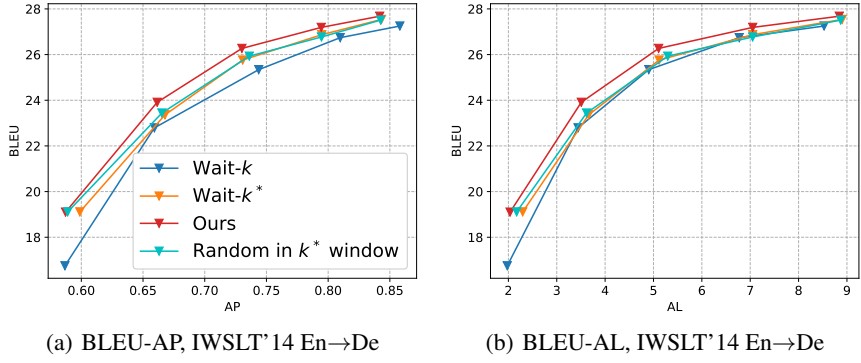

(a) BLEU-AP, IWSLT'14 En→De     (b) BLEU-AL, IWSLT'14 En→De

Figure 9: Randomly selecting $k$ in a window around $k^*$.

(3) **An annealing strategy**: Inspired by Figure 5, we design a baseline where we randomly sample the waiting threshold $m$ from a distribution $p_t(m)$ at each training step $t$. The distribution $p_t(m)$ linearly anneals from a uniform distribution to a distribution which prefers larger $m$. We expect a single annealing strategy can train reasonably good models for different inference-time $k$ values. Suppose the minimal $m$ value is $m_{\min}$, the maximal $m$ value is $m_{\max}$. $m_{\min}$ and $m_{\max}$ are two integers and $m \in \{m_{\min}, m_{\min} + 1, \cdots, m_{\max}\}$. The total training step is denote $T$. $p_t(m)$ is mathematically defined as follows:

$$p_t(m) = (1 - \frac{t}{T}) \cdot p_{\text{init}}(m) + \frac{t}{T} \cdot p_{\text{final}}(m),$$

$$p_{\text{init}}(m) = \frac{1}{m_{\max} - m_{\min} + 1}, \quad p_{\text{final}}(m) = \frac{m}{\sum_{i=m_{\min}}^{m_{\max}} i}. \tag{6}$$

The results are shown in Figure 10, which shows this baseline brings limited improvement compared to *wait-k*. A possible reason is that this baseline cannot guarantee the best "annealing" strategy for each separate $k$, while our method can adaptively find the optimal strategy. Besides, as shown in Figure 5, the learned strategies for different *wait-k* inference are pretty different.

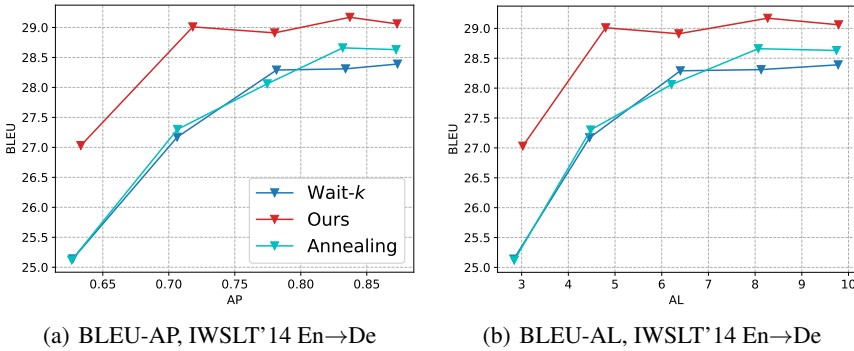

(a) BLEU-AP, IWSLT'14 En→De         (b) BLEU-AL, IWSLT'14 En→De

Figure 10: Randomly selecting $k$ in a window around $k^*$.

