# OpenReview forum: "Learning to Use Future Information in Simultaneous Translation"
_ICLR.cc/2021/Conference — Reject_

### Official Review · AnonReviewer4 · 2020-10-28
**The effectiveness of the proposed method is not convincing**

**Rating:** 5
**Confidence:** 4

**Review:**

This paper improves the wait-k based simultaneous NMT by training on an adaptive wait-m policy. The proposed method and experiments are clearly described. Experiments demonstrate that the proposed method is significant better than the wait-k baseline.

However, compared to heuristic-based baselines, seemingly the proposed method is not significantly better in most cases, especially on the WMT En-De dataset. Given that the WMT dataset is much larger than IWSLT datasets, does this suggest that the proposed method may not work well on larger dataset?

I’m also curious about the reason of using adaptive wait-m only during training since it is a more straightforward idea to apply adaptive policy during both training and inference. Would it be better if the adaptive policy (by re-design features) was used on inference as well?

---

> ### Author Response · Authors · 2020-11-24
> **Comments for AnonReviewer4**
>
> > [Q1] method is not significantly better in most cases, especially on the WMT En-De dataset. Given that the WMT dataset is much larger than IWSLT datasets, does this suggest that the proposed method may not work well on larger dataset?
>
> We agree that that gaps between our method and wait-k on WMT dataset are not so large as those for IWSLT datasets, but our method is still consistently better than all baselines. In the new version, we include the results of all heuristic baselines on WMT dataset in Figure 3, and find that our method consistently outperform heuristic baselines. We will keep working on larger datasets.
>
> > [Q2] I'm also curious about the reason of using adaptive wait-m only during training since it is a more straight forward idea to apply adaptive policy during both training and inference. Would it be better if th eadaptive policy (by redesign features) was used on inference as well?
>
> Thanks for your suggestions, and we will redesign the features in the future version to get an RL-based adaptive inference policy. Currently, we can extend our model to an adaptive inference strategy to further boost the performance. Please kindly check the
> "[General reply](https://openreview.net/forum?id=YjXnezbeCwG&noteId=zng-7L28CR8)" for more details.

---

### Official Review · AnonReviewer3 · 2020-10-28
**Using a sledgehammer to kill a fly?**

**Rating:** 5
**Confidence:** 4

**Review:**

This paper proposes a new training method for wait-k simultaneous translation. Rather than training on prefix pairs where the target prefix lags the source by k tokens, it uses an RL controller to determine an optimal lag for each sentence pair. The controller uses a small set of features intended to capture training progress, and is trained with REINFORCE to minimize wait-k loss on a validation set, in alternation with main training steps. This method shows consistent gains over various wait-k training heuristics, and some gains over other approaches that adapt the lag at inference time.

The paper is very well written and organized. The method makes sense, and the experiments are quite thorough, comparing to a competitive set of heuristic baselines, and showing credible - though fairly modest - gains in this setting.

The comparisons to adaptive baselines are less convincing. They are shown only for one small-data language pair, no implementation details are given (for instance, architectures and model capacities), and the relative results are very different from those in the literature (MILK << wait-k, WIW, WID). I think the paper would be stronger if these were simply omitted.

Given its ease of implementation and efficiency, I think there is room for a paper focusing on improving wait-k training, even if wait-k isn’t quite state of the art. But having to set up an RL controller detracts from this picture, especially since the gains over various heuristics - in particular the random heuristic that works for any inference-time k - aren’t spectacular. My main reaction to this work is that it should be possible to get most of the gains using some predetermined curriculum inspired by figure 5 (random sampling at the beginning, annealing to some m > k) that is effective for a broad range of inference-time k’s. Note that this is quite different from the CL baseline included in the results. As it stands, I fear this paper risks being a dead end: too complex to be worth implementing, as the field moves on past wait-k.

Questions and suggestions:

1. Controller feature (6): will this value ever repeat during an entire training run?
2. For Mk, since this runs only over the validation set and you are using RL, why not use the actual wait-k BLEU score?
3. You need to include all heuristic baselines in figure 4.

---

> ### Author Response · Authors · 2020-11-24
> **Comments for AnonReviewer3**
>
> > [Q1] The comparisons to adaptive baselines are less convincing. They are shown only for one small-data language pair, no implementation details are given (for instance, architectures and model capacities), and the relative resultsare very different from those in the literature (MILK << wait-k, WIW,WID). I think the paper would be stronger if these were simply omitted.
>
> Sorry that the comparison is confusing. For MMA and MILk, we directly use the results reported in [REF 1], and the results for MILk are reproduced results by [REF 1]. For WIW and WID, we re-implement these algorithms using the exact same model architecture and training setting as in [REF 1]. We have listed the implementation details in Appendix B.3. We are not sure why the results for MILk are that low. We will reproduce MMA and MILk on several more dataset to get more comparisons.
>
> > [Q2] My main reaction to this work is that it should be possible to get most of the gains using some predetermined curriculum inspired by figure 5 (random sampling at the beginning, annealing to some m>k) that is effective for a broad range of inference-time k's.
>
> Thanks for your suggestion. We conduct experiments on this idea, and find that this idea does not bring much improvement compared to wait-$k$. The detailed settings and results are repoted in Appendix D.3 (see the paragraph startswith "an annealing strategy"). A possible reason is that this baseline cannot guarantee the best "annealing" strategy for each separate k, while our method can adaptively find the optimal strategy. Also from Figure 5, we can find that the learned strategy for different inference-time k’s are pretty different.
>
> > [Q3] Controller feature (6): will this value ever repeat during an entire training run?
>
> It is not repeated.  Besides, as in Table 4, we find that removing this feature causes the performance to drop (see the last row of Table 4).
>
> > [Q4] For Mk, since this runs only over the validation set and you are using RL, why not use the actual wait-k BLEU score?
>
> This is because validation loss score is faster to obtain (since no decoding process is required) and easier to optimize (a smaller perturbation of the sequences might cause a significantly different BLEU score).
>
> > [Q5] You need to include all heuristic baselines in figure 4.
>
> Thanks for your suggestion. We now include all heuristic baselines in Figure 3 of the new version. The heuristic baselines do not bring much improvement over wait-$k$ on WMT'15 dataset, and our method still outperforms these baselines.
>
> **References**:
>
> [REF 1] Xutai Ma, Juan Pino, James Cross, Liezl Puzon, and Jiatao Gu. Monotonic multihead attention. In 8th International Conference on Learning Representations, 2020

---

> > ### Comment · AnonReviewer3 · 2020-11-24
> > **still difficult to motivate this work**
> >
> > Thanks for your response. I appreciate that you went to the trouble of testing out an annealing heuristic, but I'm not sure that this changes the basic picture. I'm convinced that you do better than the heuristics, but wonder whether the improvements are worth the effort, especially given the existence of more powerful adaptive methods.

---

> > > ### Author Response · Authors · 2020-11-25
> > > **Reply to "still difficult to motivate this work"**
> > >
> > > Thanks for your quick response!
> > >
> > > > Towards “whether the improvements are worth the effort”
> > > 1.	We make consistent improvement on three IWSLT tasks and one WMT task over a series of baselines (including both heuristic and adaptive baselines). The results for IWSLT tasks are available at Figure 2, Table 1 Figure 4 and Table 5. The results for the WMT task are at Figure 3, Table 6 and Figure 7.
> > > 2.	Compared to standard wait-k, our method only requires about 20%-30% time without too much hyperparameter tuning (see Table 2). In comparison, to use wait-k*, we need to train multiple wait-k models with different waiting thresholds. For CL, we need to carefully design the annealing strategy. Both wait-k* and CL require much additional training time.
> > > 3.	Our code has been released (anonymously) for reproducibility.
> > >
> > > > Towards “especially given the existence of more powerful adaptive methods”
> > >
> > > 1.	We compare with five adaptive methods, including WIW, WID, MILk, MMA-IL and MMA-H on IWSLT En$\to$Vi translation. The results are shown in the Figure 4(a) of the manuscript.  We can see that when AL $\ge 5.0$, our method outperforms all baseline models, and when AL < 5.0, our method performs slightly worse than MMA-IL and MMA-H. That is, with our proposed method, wait-k could surpass the performance of many adaptive baselines. We will consider applying our method directly to adaptive algorithms in the future.
> > > 2.	Recently, [REF 1] proposed a method that can adaptively leverage a set of wait-k models for decoding.  We make a combination with this method and obtain further improvement.  Please kindly re-check the [General reply: combination with adaptive decoding/inference](https://openreview.net/forum?id=YjXnezbeCwG&noteId=zng-7L28CR8).
> > >
> > > In summary,
> > >
> > > 1.	On IWSLT En$\to$Vi, our method generally achieves better results than adaptive methods like WIW, WID, MILk, MMA-IL and MMA-H.
> > > 2.	We make a first step on combing our method with an adaptive decoding strategy [REF 1] and obtain more improvement. We will combine our method with more adaptive algorithms in the future.
> > >
> > > **References**
> > >
> > > [REF 1] Baigong Zheng, Kaibo Liu, Renjie Zheng, Mingbo Ma,Hairong Liu, and Liang Huang. Simultaneous translation policies: From fixed to adaptive. In Proceedings of the 58th Annual Meeting of the Association forComputational Linguistics, pp. 2847–2853, 2020a. doi: 10.18653/v1/2020. acl-main.254. URL https://www.aclweb.org/anthology/2020.acl-main.254

---

### Official Review · AnonReviewer2 · 2020-10-28
**Promising methods, but suspicious settings of model training**

**Rating:** 4
**Confidence:** 4

**Review:**

This paper proposes a training strategy for simultaneous translation to choose appropriate amount of look-ahead information for each decoding. Based on the observation that the wait-k method can be improved by training with longer information, the method introduces a function to determine its length given the current example (source x, target y, and the translation model f). It is used as only a guidance during training, and it remains the same decoding criterion at inference.

The method is interesting and has promising improvements compared with bare wait-k methods according to the experiments, but the paper seems to have some major questions which should affect the conclusion. I recommend to revise the paper appropriately, especially to resolve following concerns:

- The proposed method is intuitively strange because of mismatching between training/inference strategies. It is also unclear to choose this method rather than adaptive wait-k methods, e.g., one referred as Zheng et al. (2020a), which may solve a similar problem directly. The paper needs at least some comparison of these kind of methods to figure out the advantages of the proposed method.
- Algorithm 1 involves a suspicious use of the development set: it is used directly to optimize a parameter. Specifically, since \omega is optimized using the D_va, it brings information of the D_va into f, resulting that the training process does not include any strategy to avoid overfitting (in other words, your training set is actually D_tr + D_va, and there is no so-called development set).

Minor comments:

- The title sounds misleading: the proposed method still does not learn how to use future information because it uses only k look-ahead information at inference (same as usual wait-k methods), i.e., there is nothing special to represent "future".
- Figure 1 involves some common mistakes of using bar charts: it must use 0 as the origin, and must not shorten the bar. If you want to focus on differences between each value, you should use other chart.

---

> ### Author Response · Authors · 2020-11-24
> **Comments for AnonReviewer2**
>
> Thanks for your review comments.
>
> > [Q1] The proposed method is intuitively strange because of mismatching between training/inference strategies.
>
> We are aware that there is training/inference mismatch in our method. Despite the mismatch, our method still brings much benefits, and the reason is as follows: To get a perfect translation, we should use the entire source sentence as input, which is not possible in simultaneous translation during inference. Access to future information reduces the gap of information between perfect translation and simultaneous translation, and improves training performance. Thus, although the training/test mismatch might increase the generalization error, we improve the final translation quality, as shown by the experiments.
>
> > [Q2] It is also unclear to choose this method rather than adaptive wait-k methods, e.g., one referred as Zheng et al. (2020a), which may solve a similar problem directly. The paper needs at least some comparison of these kind of methods to figure out the advantages of the proposed method.
>
> Following your suggestion, we conduct some experiments with Zheng et al. (2020a). Please kindly check "[General reply](https://openreview.net/forum?id=YjXnezbeCwG&noteId=zng-7L28CR8)" for details.
>
> > [Q3] Algorithm 1 involves a suspicious use of the development set...
>
> Leveraging development set to improve the models is widely used in meta-learning literature [REF1, REF2] and neural architecture search literature [REF3, REF4]. The objective function in these works is usually formulated as a bi-level optimization, where we want to minimize the validation loss of model $\theta^*$ (on the dev set), and $\theta^*$ is obtained on the training corpus.
>
> > [Q4] The title sounds misleading: the proposed method still does not learn how to use future information because it uses only k look-ahead information at inference (same as usual wait-k methods), i.e., there is nothing special to represent "future".
>
> The "future information" is utilized in the training process rather than inference process. In this paper, we want to improve wait-$k$ without introducing additional cost during inference stage. At training time, to obtain a better wait-$k$, we introduce an RL controller to adaptively provide future information (i.e., $m$ words ahead where $m>k$) and eventually obtain better results. Sorry for the misleading. After the anonymous period, we will change the title.
>
> > [Q5] Figure 1 involves some common mistakes of using bar charts: it must use 0 as the origin, and must not shorten the bar.
>
> Sorry that the figure is misleading. Please check out new Figure 1.
>
> **References**:
>
> [REF 1] Learning to Teach, https://arxiv.org/abs/1805.03643
>
> [REF 2] Learning to Teach with Dynamic Loss Functions, https://arxiv.org/abs/1810.12081
>
> [REF 3] DARTS: Differentiable Architecture Search, https://arxiv.org/abs/1806.09055
>
> [REF 4] Efficient Neural Architecture Search via Parameter Sharing, https://arxiv.org/abs/1802.03268

---

### Official Review · AnonReviewer1 · 2020-10-28
**complicated method with marginal improvements**

**Rating:** 5
**Confidence:** 4

**Review:**

The authors observed that some lookahead information during training time is helpful to improve the translation accuracy for simultaneous translation. Base on this observation, this paper proposes to use RL-based methods to learn a certain number of lookahead words during the training of the wait-k-based simultaneous translation model.

This paper proposes a new approach for improving the translation quality and the results indeed show some improvements over the baseline methods. However, I still have the following concerns:
1) I think the proposed RL-based methods are very completed (in terms of hyperparameter searching, extra training time compared with other baselines) but the improvements are quite marginal compared with wait-k* and random.

2) For the experiments, the authors did not compare with other agent-based or adaptive methods.

3) This paper only designs a controller for training. I think we could also have a controller for the inference. In this way, I believe we could use regular wait-k training, and use a controller to decide a smaller k during inference. Then it will be very similar to Gu. et al's RL-based methods. I suggest the authors include this method's experiments as well.

4) If I did not misunderstand, the m of wait-m is defined on each training pair, but I think this will dramatically increase the training time and hard to do batch training. How slow is your training compared with baseline wait-k? I believe the baseline wait-k is already very slow.

---

> ### Author Response · Authors · 2020-11-24
> **Comments for AnonReviewer1**
>
> Thanks for your review comments.
>
> > [Q1] the improvements are quite marginal comparedwith wait-k* and random…
>
> We respectively disagree with the claim that the improvements are marginal. As shown in Table 5 (in Appendix C), in most cases, our method outperforms the heuristic baselines including wait-$k^*$, CL and random by 0.5-1.5 points. In addition, the improvements brought by heuristic baselines are inconsistent under different language pairs and different $k$ values, while our method brings more consistent improvements.
>
> > [Q2] For the experiments, the authors did not compare with other agent-based or adaptive methods.
>
> We did compare our method with two agent-based methods (wait-if-diff and wait-if-worse [REF 1]) and two adaptive methods (MMA [REF 2] and MILk [REF 3]), and the results are shown in Figure 4(a) and Figure 6(a). Our method outperforms most baselines, except that our method performs similarly with MMA under low latency.
>
> > [Q3] This paper only designs a controller for training. I think we could also have a controller for the inference. … Then it will be very similar to Gu. et al's RL-based methods.
>
> Thanks for the suggestion. In this work, our focus is to leverage future information to improve the wait-k inference. We will design a controller for inference in the future.
>
> 1. We try to implement Gu et al’s RL method (i.e.,[REF 4]), but due to time limitation, we were not able to get reasonable results. The main difficulty is that using the adaptive strategy makes it hard to conduct batch inference, and thus makes the model extremely slow. Besides, Gu et al’s is outperformed by wait-k [REF 5] (please check Figure 5~8 of [REF 5]). A possible explanation is that simply optimizing the inference stage is not enough, and optimizing the training process is crucial.
> 2. Alternatively, we can combine our method with another adaptive decoding strategy. Please refer to "[General reply](https://openreview.net/forum?id=YjXnezbeCwG&noteId=zng-7L28CR8)".
>
> > [Q4] the m of wait-m is defined on each training pair, but I think this will dramatically increase the training time and hard todo batch training
>
> We empirically verify that our method does not dramatically increase the training time. We record the training speed of wait-k and our methods on IWSLT datasets, and our method requires 20% - 30% additional training time. This additional calculation is mainly brought by calculating training loss as part of the input feature $\varphi$. If we remove the training loss feature, our method only requires ~5% additional training time. The detailed results are reported in Section 5.3.
>
> For implementation, wait-$m$ training strategy is implemented by masking the encoder-decoder attention. To use a different $m$ for each different data pair, we just need to create a different mask for each data pair, where the masks are of the same shape and can be easily batched.
>
> **References**:
>
> [REF 1] Kyunghyun Cho and Masha Esipova. Can neural machine translation do simultaneous translation? CoRR, abs/1606.02012, 2016. URL http://arxiv.org/abs/1606.02012.
>
> [REF 2] Xutai Ma, Juan Pino, James Cross, Liezl Puzon, and Jiatao Gu. Monotonic multihead attention. In 8th International Conference on Learning Representations, 2020
>
> [REF 3] Naveen Arivazhagan, Colin Cherry, Wolfgang Macherey, Chung-Cheng Chiu, Semih Yavuz, Ruoming Pang, Wei Li, and Colin Raffel. Monotonic infinite lookback attention for simultaneous machine translation. In Proceedings of the 57th Annual Meeting of the Association for Computational Linguistics, pp. 1313–1323, Florence, Italy, 2019. Association for Computational Linguistics. URL https://www.aclweb.org/anthology/P19-1126.
>
> [REF 4] Jiatao Gu, Graham Neubig, Kyunghyun Cho, and Victor O.K. Li. Learning to translate in real time with neural machine translation. In Proceedings of the 15th Conference of the European Chapter of the Association for Computational Linguistics: Volume 1, Long Papers, pp. 1053– 1062, Valencia, Spain, April 2017. Association for Computational Linguistics. URL https://www.aclweb.org/anthology/E17-1099.
>
> [REF 5] STACL: Simultaneous Translation with Implicit Anticipation and Controllable Latency using Prefix-to-Prefix Framework, https://www.aclweb.org/anthology/P19-1289.pdf

---

### Author Response · Authors · 2020-11-24
**General reply: combination with adaptive decoding/inference**

We extend our method to an adaptive version by combining with [REF 1], which conducts adaptive decoding by combining a set of standarad wait-$m$ models. More specifically, to use [REF 1], we first prepare a set of pre-trained wait-$m$ models with different $m$ values, and then adaptively select the waiting threshold during inference. Therefore, we can combine [REF 1] with our method, where the wait-$m$ models are obtained through our strategy.

We conduct experiments on IWSLT En$\to$Vi and have the following observations: (1) our method outperforms [REF 1]; (2) after combing our approach with [REF 1], the performance can be further improved, which shows that our method is complementary to adaptive inference strategies like [REF 1].

More details can be found at the last paragraph of Section 4.2, the last paragraph of Section 5.2, and Figure 4(b).

What we want to emphasize is that, to use the method in [REF1], we need to prepare a set of wait-k models in advance, which increase the training cost. Besides, the inference process will be significantly slower, as we need to decode using multiple models simultaneously. In comparison, our method focuses on how to improve a single wait-k strategy without introducing additional inference cost, and the additional computational overhead in training process is acceptable (see Section 5.3).

**References**:

[REF 1] Baigong Zheng, Kaibo Liu, Renjie Zheng, Mingbo Ma,Hairong Liu, and Liang Huang. Simultaneous translation policies: From fixed to adaptive. In Proceedings of the 58th Annual Meeting of the Association forComputational Linguistics, pp. 2847–2853, 2020a. doi: 10.18653/v1/2020.
acl-main.254. URL https://www.aclweb.org/anthology/2020.acl-main.254

---

### Decision · Program_Chairs · 2021-01-07
**Final Decision**

**Decision:**

Reject

**Comment:**

This paper improves the wait-k based simultaneous NMT by training on an adaptive wait-m policy with a controller determining the lag for sentence pair.  The controller is trained with RL to minimize the loss on a validation set. The overall model is reasonable, which is well presented. I however have the following two concerns
1. There is a clear mismatch between training/inference strategies, which raises two problems
    1. The motivation:  the authors tried to explain that in discussion,  but it is not convincing enough
    2. The title is misleading since there is no future information to use during inference
2. The experiments is not convincing enough in that a) the improvement over baseline is modest, and b) comparison to adaptive wait-k and other strong baseline is insufficient

In conclusion I would suggest to reject this paper.